# BiF³-Net: A Full BiFormer Full-scale Fusion Network for Accurate Gastrointestinal Images Segmentation

**Yunze Wang**[*1]  YUNZE.WANG19@STUDENT.XJTLU.EDU.CN
**Silin Chen**[*2]  19271205@BJTU.EDU.CN
**Xi Long**[1]  XI.LOONG@OUTLOOK.COM
**Yi Tian**[1]  YI.TIAN21@STUDENT.XJTLU.EDU.CN
**Ye Huang**[1]  YE.HUANG21@STUDENT.XJTLU.EDU.CN
**Tianyang Wang**[1]  TIANYANG.WANG21@STUDENT.XJTLU.EDU.CN
**Jingxin Liu**[1]  JINGXIN.LIU@XJTLU.EDU.CN

[1] *Xi'an Jiaotong - Liverpool University*
[2] *Beijing Jiaotong University*

## Abstract

UNet-like segmentation models have been widely explored for the computer-aided segmentation and diagnosis of gastrointestinal (GI) tract diseases. However, the UNet architecture encounters two primary challenges: limited receptive fields due to conventional convolution operations, and a semantic gap arising from simplistic skip connections. In this paper, we introduce **BiF³-Net**, a novel model that integrates BiFormer blocks throughout the encoder and decoder, along with a full-scale BiFormer Fusion Bridge (BFB) module, aimed at addressing the aforementioned limitations. Meanwhile, we propose the Dense Inception Classifier (DIC) module to mitigate the over-segmentation problem in non-organ images. Extensive experiments demonstrate the effectiveness and adaptability of the proposed model.

**Keywords:** Content-aware Sparse Attention, Full-scale Fusion, Over-Segmentation

## 1. Introduction

Gastrointestinal (GI) cancer is a malignant tumor affecting the digestive organs, accounting for 26% of the global cancer incidence and 35% of all cancer-related deaths (Arnold et al., 2020). Accurate segmentation of gastrointestinal tumors is crucial for subsequent radiotherapy. However, manual delineation is a tedious and labor-intensive task.

UNet (Ronneberger et al., 2015), as a hierarchical encoder-decoder fully convolutional architecture with skip connections, has shown promising performance in automated medical image segmentation. However, its limited receptive field due to traditional convolution operations hinders its effectiveness. ViT (Dosovitskiy et al., 2020), benefiting from capturing long-range dependencies, has achieved outstanding performance in natural image classification. Nevertheless, its architecture primarily faces two challenges: first, traditional ViT architectures use columnar structures to input images, utilizing only single-scale feature maps. Second, ViT's computational complexity scales quadratically with image size. These drawbacks limit ViT's further application in dense tasks such as medical image segmentation. A series of works have introduced progressive hierarchical structures and inductive biases to reduce attention computation, such as Swin-UNet (Cao et al., 2022), Medical Transformer (Valanarasu et al., 2021), respectively restricting attention operation within local windows, axial stripes. However, these attention mechanisms are based on manually designed patterns. In this paper, as illustrated in Figure 1(a), we employ a consecutive content-aware sparse

---

[*] Contributed equally

attention block, BiFormer (Zhu et al., 2023), to construct our hierarchical UNet-like BiF³-Net. Specifically, in the BiFormer block, irrelevant key-value pairs are first filtered out at a coarse region level for a query, followed by fine-grained token-to-token attention applied in the union of remaining candidate regions. Since simple skip connections in UNet-like networks merely copy features from the encoder to the decoder, which may not sufficiently bridge the semantic gap between them, we design the BiFormer fusion bridge, as shown in Figure 1(b), to alleviate this gap. In addition, considering non-organ and over-segmentation phenomena in medical images, we design a Dense Inception Classifier (DIC) module, a class of Inception structure used to determine whether an image needs segmentation. Only those classified as requiring further segmentation will undergo loss computation.

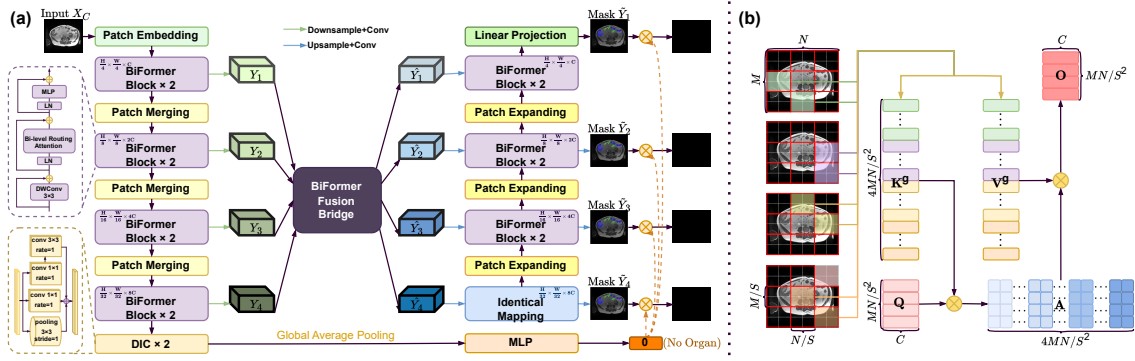

Figure 1: **(a)** The overview of our proposed framework BiF³-Net for gastrointestinal image segmentation. **(b)** The details of the BiFormer Fusion Bridge module.

## 2. Methods

**Backbone.** The BiF³-Net employs fully BiFormer-based encoders and decoders. In the encoder, input data is processed through BiFormer blocks and patch merging modules to form hierarchical features. Subsequently, in the decoder, features are upsampled to produce mask output using BiFormer blocks and patch expanding modules. The BiFormer block serves as the basic unit of BiF³-Net. Specifically, given a $2D$ input, BiFormer first divides the input feature map into non-overlapping patches of size $S \times S$ and derives the query, key, and value $Q, K, V \in \mathbb{R}^{S^2 \times \frac{H \times W}{S^2} \times C}$. Then, we derive region-level queries and keys $Q_r, K_r \in \mathbb{R}^{S^2 \times C}$, by applying per-region averaging on $Q$ and $K$ (note: each region contains multiple patches). We multiply $Q_r$ with $K_r$ to obtain $A_r$, measuring how much two regions are semantically related. Then, we use a row-wise *top-k* operator to obtain the *top-k* most relevant regions for each region. Finally, patches within each region only interact with patches within the corresponding *top-k* most relevant regions through standard attention mechanisms.

**BiFormer Fusion Bridge.** The BiFormer fusion bridge (BFB) module addresses the semantic gap between the encoder and decoder by extending the attention interaction within a single feature map, akin to the BiFormer module. Additionally, it enables interaction across different stages of the encoder by computing the *top-k* most relevant regions within each feature map and performing basic attention operations between these selected regions, as shown in Figure 1(b). To maintain consistency with BiFormer, other settings are preserved. As our framework is hierarchically designed, feature map sizes vary across different levels in

the encoder and decoder. To address this, adaptive pooling and $1\times1$ depth-wise convolutional operations are applied to ensure uniform feature map sizes across all stages in the encoder. **Dense Inception Classifier.** The classification branch DIC, aimed at alleviating over-segmentation, utilizes the output of the final stage of the encoder as input and performs classification through a network consisting of several parallel pathways and a MLP layer. Specifically, our DIC module comprises a $3\times3$ maximum pooling layer, a $1\times1$ convolutional layer (CL), and $1\times1$ and $3\times3$ CLs with an atrous rate of 1.

## 3. Experiments and Conclusion

**Dataset.** The UW-Madison GI Tract Segmentation dataset (happyharrycn, 2022) is used in this study, which comprises 38,496 MRI slices from 85 cases, with 21,906 annotated for the large bowel, small bowel, and stomach, and the rest as background. The dataset is randomly split into 80% for training and 20% for testing, 10% of the training set is used for validation. **Setting.** All images are resized to $224\times224$, random flip and crop is used for data augmentation. The initial learning rate is set to $2e^{-4}$, with a weight decay of 0.05, and a cosine schedule with warm-ups. The number of epochs is 120 with batch sizes of 16. Weighted deep supervision (WDS) is used during the training stage. The loss function is expressed as below, where $Y$ is the ground truth, $\tilde{Y}_i$ refers to mask output in different decoder stages and $\alpha_i$ refers to the weight of the deep supervision in different decoder stages.

$$\mathcal{L}_{WDS} = \sum_{i=1}^{4} \alpha_i(\mathcal{L}_{\text{Dice}}(\tilde{Y}_i, Y) + \mathcal{L}_{\text{CE}}(\tilde{Y}_i, Y)), \alpha_i = \begin{cases} 0.7 & \text{if } i=1, \\ 0.1 & \text{otherwise.} \end{cases}$$

**Results.** As shown in Table 1, we report the quantitative results of the proposed model with a series of state-of-the-art (SOTA) medical image segmentation methods on the Dice score (Dice), Intersection over-union (IoU) and Hausdorff distance (HD). Additionally, we report the number of parameters and inference time of each method. Our model outperforms other SOTA methods across the mentioned three kind of evaluation metrics with relatively fewer parameters and faster inference time, without utilizing any ImageNet pre-trained weights.

Table 1: Performance comparison and ablation study on the UW-Madison GI dataset.

| | # Params(M) | Inference Time(S) | Average | | | Large bowel | | | Small bowel | | | Stomach | | |
|---|---|---|---|---|---|---|---|---|---|---|---|---|---|---|
| | | | Dice (%) ↑ | IoU (%) ↑ | HD ↓ | Dice (%) ↑ | IoU (%) ↑ | HD ↓ | Dice (%) ↑ | IoU (%) ↑ | HD ↓ | Dice (%) ↑ | IoU (%) ↑ | HD ↓ |
| UNet(Ronneberger et al., 2015) | **24.56** | 3.12 | 87.96 | 85.01 | 1.29 | 86.08 | 82.53 | 1.54 | 85.56 | 81.89 | 1.58 | 92.77 | 90.99 | 0.68 |
| UNet3+(Huang et al., 2020) | 26.97 | 4.32 | 88.49 | 85.41 | 1.22 | 86.93 | 83.03 | 1.49 | 85.60 | 82.00 | 1.53 | 92.94 | 91.21 | 0.66 |
| Deeplabv3+(Oktay et al., 2018) | 31.23 | 4.12 | 88.61 | 85.48 | 1.21 | 87.45 | 83.45 | 1.47 | 85.69 | 82.03 | 1.49 | 92.71 | 90.97 | 0.66 |
| TransUnet(Petit et al., 2021) | 105.3 | 4.87 | 88.30 | 85.59 | 1.28 | 87.66 | 83.83 | 1.51 | 86.71 | 83.13 | 1.57 | 91.53 | 89.82 | 0.76 |
| Swin-UNet(Cao et al., 2022) | 41.40 | 3.58 | 88.56 | 85.86 | 1.21 | 87.94 | 84.31 | 1.48 | 86.66 | 83.03 | 1.59 | 92.07 | 90.13 | 0.70 |
| UCTransNet(Wang et al., 2022) | 56.13 | 5.12 | 88.96 | 86.39 | 1.17 | 88.21 | 84.48 | 1.43 | 86.95 | 83.84 | 1.51 | 91.80 | 90.21 | 0.71 |
| nn-UNet(Isensee et al., 2021) | 79.65 | 9.65 | 89.39 | 86.94 | 1.10 | 88.34 | 85.32 | 1.40 | 88.10 | 85.21 | 1.43 | 93.03 | 91.29 | 0.63 |
| Ours (Baseline) | 24.67 | **2.87** | 88.67 | 85.98 | 1.19 | 88.30 | 84.46 | 1.46 | 86.91 | 83.74 | 1.55 | 92.18 | 90.35 | 0.68 |
| Ours (w BFB) | 32.89 | 3.23 | 89.57 | 87.36 | 1.07 | 88.63 | 85.64 | 1.34 | 88.36 | 85.30 | 1.42 | 93.20 | 91.33 | 0.61 |
| Ours (w BFB & DIC) | 33.23 | 3.40 | 90.41 | 88.21 | 1.04 | 89.32 | 86.32 | 1.25 | 89.23 | 86.09 | 1.39 | 94.12 | 92.20 | 0.55 |
| **Ours (w BFB & DIC & WDS)** | 33.23 | 3.40 | **90.98** | **88.71** | **1.02** | **89.71** | **86.66** | **1.20** | **89.70** | **86.49** | **1.31** | **94.85** | **92.51** | **0.50** |

**Conclusion.** In this paper, we introduce **BiF$^3$-Net**, a comprehensive UNet variant designed for accurate segmentation of the gastrointestinal tract on MRI images. Using BiFormer sparse attention blocks, our model captures long-range dependencies and employs full-scale feature maps for improved precision. While the Dense Inception Classifier module and weighted deep supervision strategy further enhance performance. To the best of our knowledge, this is the first fully BiFormer-based medical image segmentation model. Extensive experiments and ablation studies demonstrate the effectiveness and applicability of the proposed model.

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
