# OpenReview forum: "BiF³-Net: A Full BiFormer Full-scale Fusion Network for Accurate Gastrointestinal Images Segmentation"
_MIDL.io/2024/Short_Papers — MIDL 2024 Short Papers_

### Official Review · Reviewer_kV8K · 2024-04-22

**Confidence:** 4
**Final Rating:** 4

**Review:**

Summary:
The paper introduces BiF3-Net for segmenting gastrointestinal tract images. It improves upon the traditional UNet architecture by integrating BiFormer blocks to capture long-range dependencies within images, employing a BiFormer Fusion Bridge to bridge the semantic gap between the encoder and decoder, and introducing a Dense Inception Classifier to reduce over-segmentation. Tested on a comprehensive GI tract dataset, BiF3-Net demonstrates superior precision and computational efficiency compared to current state-of-the-art methods.

Pros:

1.	The introduction of the BFB and DIC modules innovatively addresses the semantic gap and over-segmentation problems, respectively, which are common issues in similar architectures.
2.	Despite its complexity, the network maintains lower inference times and requires fewer parameters than comparable models, making it more suitable for real-time applications.
3.	The experimental results clearly show the efficacy of each introduced modules and the superior performance of our method compared to other state-of-the-art methods.

Cons:

1.	While the model performs well on GI tract segmentation, its adaptability to other segmentation tasks with different modalities is not validated.
2.	The impact of the hyperparameter 'k' in the 'top-k selected regions' on the efficacy and segmentation performance of the algorithm has not been discussed

---

### Decision · Program_Chairs · 2024-04-26

Accept